# An AI-based Framework for Diagnostic Quality Assessment of Ankle Radiographs

**Dominik Mairhöfer**[1]          MAIRHOEFER@INB.UNI-LUEBECK.DE
**Manuel Laufer**[1,2]            LAUFER@INB.UNI-LUEBECK.DE
**Paul M. Simon**[3]           PAULMARTIN.SIMON@UKSH.DE
**Malte Sieren**[3]            MALTE.SIEREN@UKSH.DE
**Arpad Bischof**[3,4]           ARPAD.BISCHOF@UKSH.DE
**Thomas Käster**[2]            TK@PRCMAIL.DE
**Erhardt Barth**[1]           BARTH@INB.UNI-LUEBECK.DE
**Jörg Barkhausen**[3]         JOERG.BARKHAUSEN@UKSH.DE
**Thomas Martinetz**[1]       MARTINETZ@INB.UNI-LUEBECK.DE

[1] *Institute for Neuro- and Bioinformatics, University of Lübeck, Germany*

[2] *Pattern Recognition Company GmbH, Lübeck, Germany*

[3] *University Medical Center Schleswig-Holstein, Lübeck, Germany*

[4] *IMAGE Information Systems Europe GmbH, Rostock, Germany*

## Abstract

The quality of radiographs is of major importance for diagnosis and treatment planning. While most research regarding automated radiograph quality assessment uses technical features such as noise or contrast, we propose to use anatomical structures as more appropriate features. We show that based on such anatomical features, a modular deep-learning framework can serve as a quality control mechanism for the diagnostic quality of ankle radiographs. For evaluation, a dataset consisting of 950 ankle radiographs was collected and their quality was labeled by radiologists. We obtain an average accuracy of 94.1%, which is better than the expert radiologists are on average.

**Keywords:** radiographs, quality assessment, anatomical features, deep learning

## 1. Introduction

As one of the most frequently used imaging modalities, radiographs are of significant importance for diagnosis and treatment planning. For these tasks, a diagnostic-adequate image quality is mandatory.

Currently, radiographers have to decide if the quality of the radiograph suffices for the diagnosis or if the imaging process must be repeated. To not be able to immediately judge the diagnostic quality correctly can result in various disadvantages including unnecessary radiation exposure. Reasons for misjudging image quality as sufficient may be time pressure, inexperience or overtiredness, in which case the treating radiologist has to schedule a new examination resulting in additional effort. In the worst case, the radiographer would take a second radiograph although the first one was sufficient and thereby re-expose the patient to radiation. To prevent these errors and to establish a quality control mechanism, an automated quality assessment can help.

While there is extensive research assessing radiograph quality based on technical factors such as contrast and noise (Esses et al., 2018; Takaki et al., 2020; Wang et al., 2020), these

parameters are less important with digital radiography. A more important quality criterion, is the alignment of the body part to be X-rayed relative to the X-ray machine. A radiograph may be of perfect technical quality but can nevertheless be worthless for diagnostic purposes if relevant anatomical structures are not visible due to misalignment. To our knowledge, previous approaches do not assess the quality of a radiograph based on this criterion.

In this paper, we propose a framework based on classification and segmentation Neural Networks, which assesses the diagnostic quality of ankle radiographs based on anatomical features. Furthermore, we test the framework on a new dataset containing radiographs of ankles, with 950 radiographs in two different radiographic views (*anterior posterior* and *lateral*), all labeled by four radiologists. Using this framework, radiographers will be able to immediately get a first quality assessment of the taken radiographs without relying on a radiologist. Besides reducing the described judgment errors, the framework can be used as a quality control mechanism to detect causes for low quality radiographs.

## 2. Related Work

In recent years, deep learning has become more common in radiology (Choy et al., 2018; Saba et al., 2019). Scientists working on radiographs successfully applied Neural Networks to detect fractures (Lindsey et al., 2018; Thian et al., 2019), classify body parts (Agunwa et al., 2019), and radiographic views (Fang et al., 2020), to facilitate the work process in radiology. Although our proposed framework also includes radiographic view recognition, these steps are only part of the preprocessing for assessing diagnostic quality. Distinct to Fang et al. (2020), where only a single step is used for recognition, we use multiple steps containing different networks and resign to recognize laterality.

Esses et al. (2018) and (Wang et al., 2020) focus on automated diagnostic quality evaluation of MRT images using Neural Networks. Due to the different modalities of the imaging systems one can not easily transfer the results to radiographs.

Approaches that automatically asses the perceptual quality of radiographs only take technical parameters such as noise and contrast into account and rely on conventional computer vision methods (Samei et al., 2014; Willis et al., 2018). Takaki et al. (2020) present a deep learning approach to calculate the target exposure index for chest radiographs based on perceptual quality of small patches.

To our knowledge, there are no studies considering anatomical features for the diagnostic quality of radiographs in a deep-learning framework.

## 3. Proposed Framework

To solve the challenge of diagnostic quality assessment and to standardize the required steps, we propose a framework of several Neural Networks that is able to process radiographs of ankles and to output their diagnostic quality. It consists of the following steps: *Recognition of radiographic view*, *extraction of the region of interest (ROI)*, and *quality assessment*.

The first step relies on the fact that radiographs can be ordered hierarchically by radiographic view. A prediction of quality strongly depends on the view since corresponding criteria for radiographic views may differ. The second step prepares the input for quality assessment by removing unnecessary information.

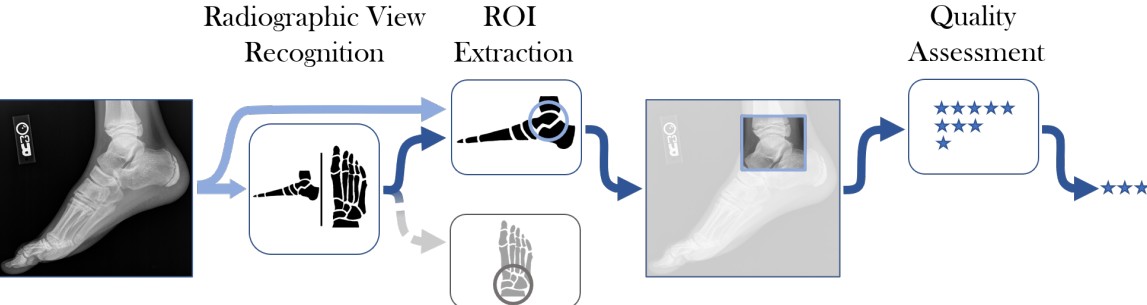

Figure 1: Schematic flow of a radiograph through the framework. For each radiograph the framework decides first which radiographic view was used. Depending on that decision the radiograph is passed to the corresponding region of interest (ROI) segmentation network. After segmentation, the resulting ROI is fed into the final quality prediction network, which outputs the quality assessment.

Each individual step can be used independently. But only within the whole framework they provide the possibility to decide whether an ankle radiograph is of high or low diagnostic quality, thereby directly supporting the radiographers in their decision process. A complete overview of the framework can be seen in Figure 1. In the following, we describe each step in more detail.

### 3.1. Recognition of the Radiographic View

The first step of the proposed framework consists of recognizing the radiographic view of the radiograph. This classification task is mandatory since radiographs of various radiographic views differ in quality assessment characteristics, as shown in Figure 2. By dividing the quality assessment task for an ankle radiograph into a view-specific task, we facilitate the learning process of our networks, since the radiographs now belong to the same domain.

### 3.2. Extraction of the ROI

While the entire radiograph is relevant for diagnosis, only a fraction is needed for assessing the quality of the standard projection (red marks in Figure 2). Based on this fact, the next step in the framework is to segment this ROI which contains the most information relevant for the diagnostic quality. An example ROI is shown in Figure 3. Since there are different quality characteristics in the radiographic views, we trained Neural Networks individually for each view. Besides removing irrelevant information, the benefit of extracting ROIs is that the subsequent quality assessment can operate on a standardized size and resolution of the relevant image part.

### 3.3. Quality Assessment

Getting standardized ROIs of a particular radiographic view is the basis for assessing the diagnostic quality with high accuracy. We use two different Neural Networks, one for each of the two radiographic views. These are trained individually on the *anterior posterior* ($AP$) and *lateral* ($LAT$) ROI, respectively and output the quality on a continuous scale from 1 to 3 (see Section 4.2).

## 4. Datasets

To test the framework presented in Section 3 two datasets were created. The first one is a collection of ankle radiographs as DICOM images and associated metadata. The second one, which to our knowledge did not exist previously in this or similar form, contains radiographs labeled by radiologists according to diagnostic quality based on anatomical features. Both datasets contain radiographs from five different X-ray machines.

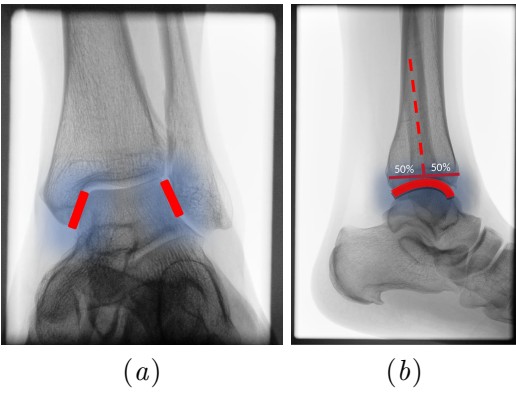

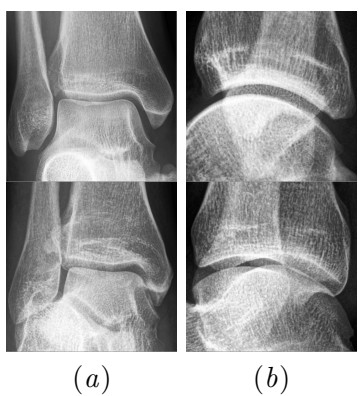

Figure 2: In $(a)$ the most relevant anatomical structures in the $AP$ radiographic view are highlighted. These include the joint gap between medial malleolus and talus as well as lateral malleolus and talus. In $(b)$ the joint space between the distal tibia and the talus is highlighted as the most relevant structure for the $LAT$ view.

Figure 3: $(a)$ shows an example ROI of a radiograph in $AP$ view with perfect alignment in the upper row and strong misalignment in the lower. $(b)$ shows the same for the $LAT$ view.

### 4.1. Weakly Labeled Dataset for Recognition of the Radiographic View

We used a dataset of 26542 ankle radiographs provided by the University Hospital Schleswig-Holstein, Campus Lübeck. From those radiographs we extracted labels for the radiographic view ($LAT$ or $AP$) with a keyword matching on the metadata. The resulting dataset contains roughly 12000 radiographs for each view. Since creating the metadata is mostly done manually and the content is not standardized, we assume that not all labels are accurate.

### 4.2. Diagnostic Quality Dataset

In order to learn the relationship between the radiographs and the quality, an annotated dataset is needed. To create such a dataset, four radiologists labeled 950 ankle radiographs, containing 475 for $LAT$ and $AP$ each.

The radiologists determined which objective criteria a radiograph of an ankle has to fulfill, to be of high diagnostic quality. One important criterion, for instance, is the complete visibility of the joint gap between medial malleolus and talus. A high diagnostic quality is a prerequisite for the radiologist to make a correct diagnosis. According to that criteria, each radiograph was labeled by each radiologist as *1* if the radiograph fulfilled the criteria perfectly, *2* if partly and *3* if the criteria were not met, and a new radiograph would have to be taken. In order to determine whether a radiograph can be used for a diagnosis, the classes *1* and *2* were grouped under the label *diagnostic* and the class *3* was labeled as *not diagnostic*. If the labels differed greatly, the radiologists had a consensus meeting. Of the $475 \cdot 4$ labels assigned for the $AP$ radiographs, 37% are *1*s, 53% *2*s and 10% *3*s. For the $LAT$ view 17% of the assigned labels are *1*s, 55% *2*s and 28% *3*s. Examples for the three classes can be seen in the Appendix in Figure 5 (a-c) for the $AP$ view and Figure 6 (a-c) for the $LAT$ view.

Additionally, each of the 950 radiographs was labeled with a ROI. As described in Section 3.2 only a fraction of the radiograph is relevant for the diagnostic quality. Therefore, the ROI was labeled as a square containing only the most relevant information. This can be seen in Figure 3. In Figure 4, which shows examples of ground truth ROI labels, it can be seen that the size of each ROI is highly dependent on the image content.

## 5. Experiments and Results

To evaluate the framework described in Section 3, each step was implemented using PyTorch and evaluated on the datasets of Section 4. To improve quality control measurements, we tested each step individually. Because of the relatively small datasets, we used the EfficientNet-B0 (Tan and Le, 2019) for classification. For segmentation a DeepLabV3 (Chen et al., 2017) with a ResNet-50 (He et al., 2016) backbone was used. Both networks were not pretrained. For all experiments we padded the input radiograph with zeros to get the desired size while maintaining the aspect ratio. Furthermore, the training radiographs were augmented with random cropping, histogram normalization, Gaussian noise, blurring, horizontal flipping and rotation. Training and test datasets were split with an 80/20 ratio.

### 5.1. Recognition of the Radiographic View

For the recognition of different radiographic views, the dataset described in Section 4.1 was used. Therefore, the last layer of the EfficientNet-B0 was modified to output two classes, either $LAT$ or $AP$, which was followed by a softmax layer to obtain class probabilities. The model was trained using the cross-entropy as loss function and stochastic gradient descent (SGD) as optimizer using a learning rate of $1 \cdot 10^{-3}$, a momentum of 0.9, a weight decay of $1 \cdot 10^{-5}$, and a batch size of 8 over 500,000 iterations. To reduce possible overfitting, the drop connect (Wan et al., 2013) rate was set to 0.4. The resulting input size of the radiographs, after augmentation, was $224 \times 224$ pixels.

Training with these parameters resulted in an accuracy of 98.4% for the test set and 98.5% for the training set. The results must be interpreted with a certain caution due to the potentially incorrectly assigned labels in the weakly labeled dataset. It may be that (i) the model predicts the correct class but the label is assigned incorrectly or that (ii) the model predicts the incorrect class and the label is also assigned incorrectly. Reviewing the resulting radiographs for case (ii) revealed 54 wrong labels for the test set and 244 for the trainings set. Taking this into account the accuracy increased to 99.5%, respectively to 99.7% for the training set. Although the actual accuracy may be slightly lower due to errors of case (i), these results clearly demonstrate that a recognition of the radiographic view can be achieved with high precision.

### 5.2. Extraction of the ROI

To segment the ROI, a DeepLabV3 was trained with the labels described in Section 4.2. The target feature map is binary, with 0 for *not ROI* and 1 for *ROI*. As segmentation output we used a single feature map, followed by a sigmoid function, to get pixel-wise outputs from 0 to 1. For the training we used the mean over the pixel wise squared error, optimized with the Adam optimizer, a learning rate of $1 \cdot 10^{-4}$, a weight decay of $1 \cdot 10^{-4}$, and a batch size of 4 over 50,000 iterations. For this task the input size after augmentation was $400 \times 400$ pixels. This training was done separately for $LAT$ and $AP$ views. Given the small dataset we used a random sub-sampling validation over 12 different dataset splits.

To measure the accuracy of the predicted ROIs the Dice score was calculated. If a pixel value of the output feature map was above a threshold of 0.7, the pixel was classified as part of the ROI. Over all 12 dataset splits the mean Dice score was 94.17% on the $AP$ views and 85.91% on the $LAT$ views. A reason for the worse result on the $LAT$ views might be that the ROIs on the $LAT$ views are significantly smaller than on the $AP$ view and thus harder to predict. Regardless of this difference in the Dice score the resulting segmentations are sufficient to get bounding boxes of the ROIs, which can be seen in Figure 4. To extract bounding boxes based on the segmentation, first the smallest fitting rectangle of the segmentation is calculated and then rotated to be horizontal. Examples with the labeled and the predicted ROIs can be seen in Figure 4.

### 5.3. Quality Assessment

For the quality assessment task an EfficientNet-B0 was used. To preserve the intrinsic order of the classes we modeled the task as a regression. One benefit of using regression is that we obtain intermediate scores. We also trained classification networks using the earth mover's distance but this led to slightly worse results. The model was trained using the mean squared error (MSE) as loss and the mean label of the four radiologist as target. The loss was minimized by SGD using a learning rate of $1 \cdot 10^{-3}$, a momentum of 0.9, a weight decay of $1 \cdot 10^{-3}$, and a batch size of 16 over 500,000 iterations. As in Section 5.1 the input size was $224 \times 224$ pixels. The same random sub-sampling validation as in Section 5.2 was used for testing.

To evaluate the accuracy of the model, an output was classified as correct if the nearest class to the continuous output was the class of the label. Evaluation on the test set resulted in a mean accuracy of 93.0% for the $AP$ view and 95.1% for the $LAT$ view, with a mean

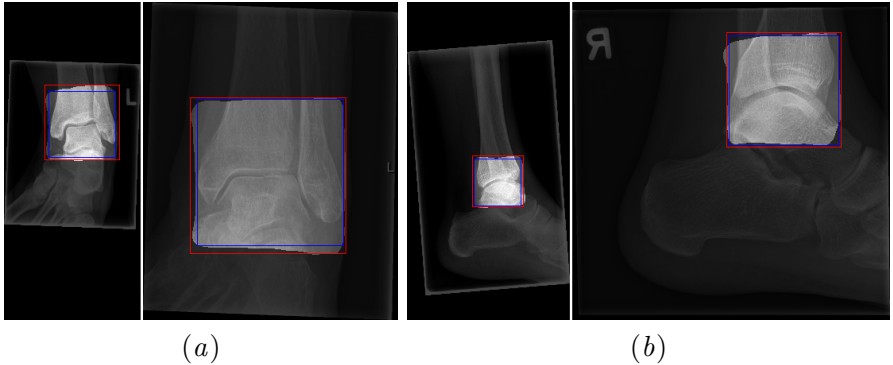

$(a)$ $(b)$

Figure 4: In $(a)$ two radiographs in the $AP$ view are shown. Their labeled ROI is marked with a blue box and the predicted ROI with a red box. The predicted segmentation mask used to construct the red box is highlighted. The same is shown in $(b)$ for the $LAT$ view. Both examples also show that the proportion of ROI in the radiograph can vary greatly.

absolute error of 0.19 for $AP$ and 0.20 for $LAT$. Over the 12 runs the standard deviation is 0.025 and 0.02 and the median accuracy 93.4% and 95.4% for $AP$ and $LAT$, respectively. The classification into *diagnostic* and *non-diagnostic* (see Section 4.2) resulted in an accuracy of 97.8% for the $AP$ view and 93.2% for the $LAT$ view. This accuracy shift is because there are different distributions of *1*s and *3*s in the $AP$ and $LAT$ parts of the dataset.

To evaluate whether the accuracy of the quality assessment benefits from the steps described in Sections 3.1 and 3.2, we repeated the training with and without these steps. The results, which are given in Table 1, show, that each step of the pipeline improves the accuracy. Overall, the mean accuracy improves from 82.4% to 94.1% when all steps are included. While the benefit of training separately for the different views is small, the extraction of ROIs seems to be necessary to obtain high accuracy. When trained without the previous view recognition, a single model is trained on the combined $AP$ and $LAT$ data to predict the quality of both views. For this each view is sampled equally often.

To get an estimation on how accurate the labels are, we tested each labeling radiologist against the others, taking one label as prediction and the mean of the remaining three as ground truth. If the difference between prediction and ground truth was at least 1, the prediction was counted as wrong. This resulted in a mean accuracy of 92.6% for $AP$ and 90.1% for $LAT$. Across the four radiologists the standard deviation is 0.026 and 0.037 for $AP$ and $LAT$, respectively. The mean accuracy over both views is 94.1% for the networks and 91.4% for the radiologists. Although our method, for evaluating the performance of the radiologists, is based on only four experts it should suffice as a first estimate.

A visual comparison of the expert labels and framework predictions on the unlabeled dataset can be seen in the Appendix in Figure 5 for the $AP$ view and Figure 6 for the $LAT$ view. For further illustration the ROIs with the highest error between expert label and predicted quality are shown in Figure 7. Note that there is no clear pattern that explains the deviation.

Table 1: Accuracy of quality assessment depending on the steps *View Recognition* (Section 3.1) and *ROI Extraction* (Section 3.2). Not training separately for *AP* and *LAT* and not extracting the ROI leads to the lowest accuracy. Both steps on their own increased the accuracy, while using both provided the best result.

Table 2: Overview of all steps in the framework and their results. The results for the *View Recognition* and the *Quality Assessment* are the achieved accuracy. For the *ROI Extraction* the result is the achieved Dice score. The *AP* and *LAT* results are not from the same model, because we trained individually for each view. Since this is not the case for the *View Recognition*, there is only a single accuracy.

| View Recog. | ROI Ext. | Accuracy | | |
|---|---|---|---|---|
| | | mean | *AP* | *LAT* |
| ✗ | ✗ | 82.4% | 80.3% | 84.5% |
| ✓ | ✗ | 85.1% | 82.9% | 87.2% |
| ✗ | ✓ | 92.4% | 92.2% | 92.5% |
| ✓ | ✓ | 94.1% | 93.0% | 95.1% |

| Step | Accuracy or Dice | | |
|---|---|---|---|
| | mean | *AP* | *LAT* |
| View Recognition | 99.5% | – | – |
| ROI Extraction | 90.1% | 94.2% | 85.9% |
| Quality Assessment | 94.1% | 93.0% | 95.1% |

## 6. Discussion

The aim of this paper was to develop a framework for automatic quality assessment and to evaluate how well it performs. We were able to show that the accuracies of the predicted quality (93.0% *anterior posterior*, 95.1% *lateral*) are better than those made by radiologists (92.6% *anterior posterior*, 90.1% *lateral*). The results of the individual steps included in the framework are summarized in Table 2. With this framework it is now possible for radiographers to immediately get a first feedback on the same level of expertise as they would get from a radiologist. These results support our view that anatomical features can be learned and are therefore suitable for the automatic assessment of diagnostic quality. In order to achieve these results, an initial separation of the radiographs into *lateral* and *anterior posterior* was necessary. This task could be achieved with an accuracy of 99.5%.

If our framework had been already in place when capturing the 950 radiographs of our dataset, 80.0% of the non-diagnostic radiographs would have been immediately and correctly recognized as such. Since 12.9% of the dataset are non-diagnostic radiographs, for every 100 radiographs the number of additional needed appointments for examinations could have been decreased from 13 to only 3.

Regarding scalability, our experiments show that about 500 labeled radiographs per radiographic view are sufficient to train a network to the accuracy level of an expert. We assume that the framework can be transferred to radiographs of other body parts. In addition to its use in day-to-day operations, the framework can potentially help to comply with quality standards and optimize the clinical routine.

## Acknowledgments

We thank Hauke Gerdes, Jan Preuß and Fabio Leal dos Reis for their help and support in collecting and labeling the radiograph datasets. This work was funded by the Bundesministerium für Wirtschaft und Energie (BMWi) through the KI-SIGS project.

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

## Appendix A. Example ROI Images

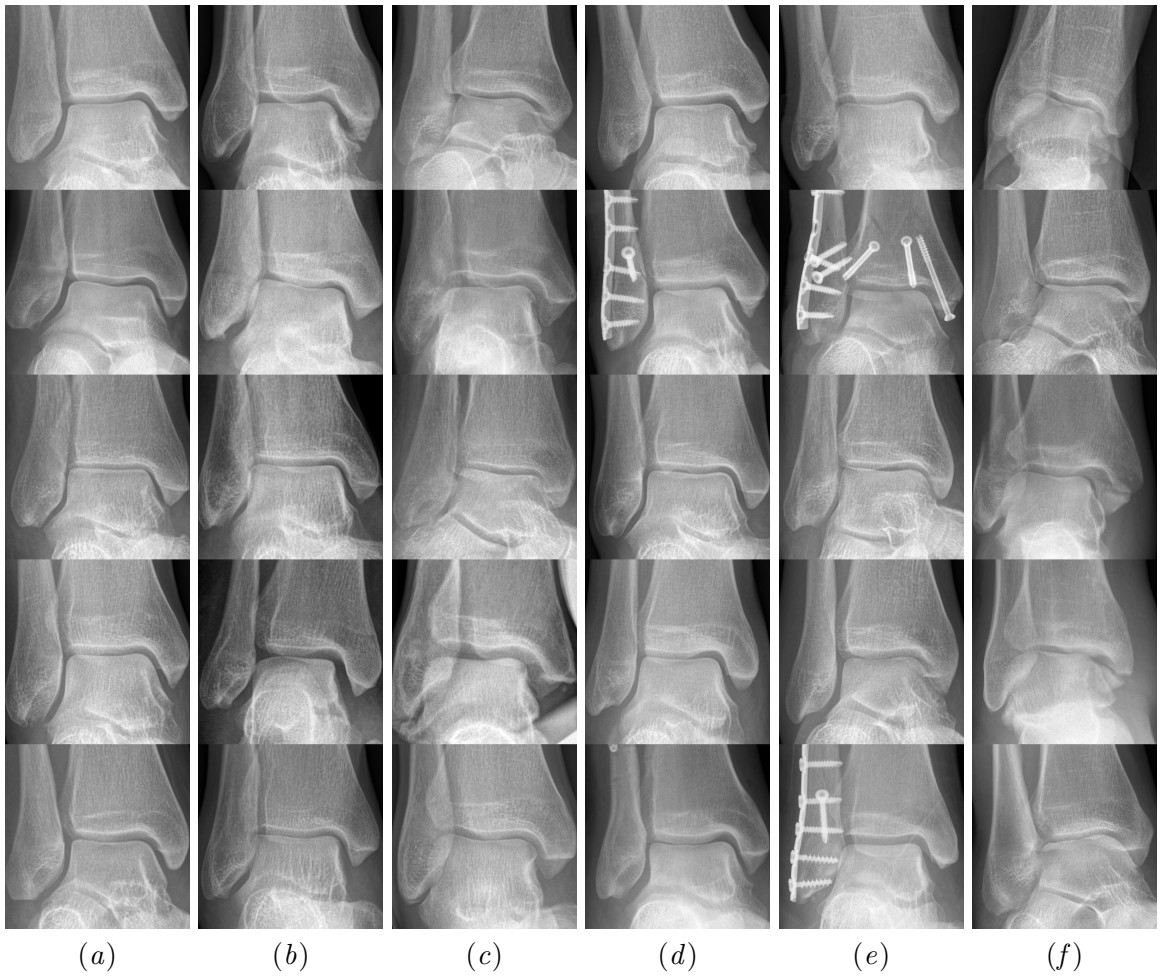

$(a)$ $(b)$ $(c)$ $(d)$ $(e)$ $(f)$

Figure 5: Each column shows five example ROIs of the labeled dataset in the *anterior posterior* view with the expert label $1(a)$, $2(b)$, and $3(c)$; and five examples of unlabeled ROIs for which our framework predicts the quality classes $1(d)$, $2(e)$, and $3(f)$.

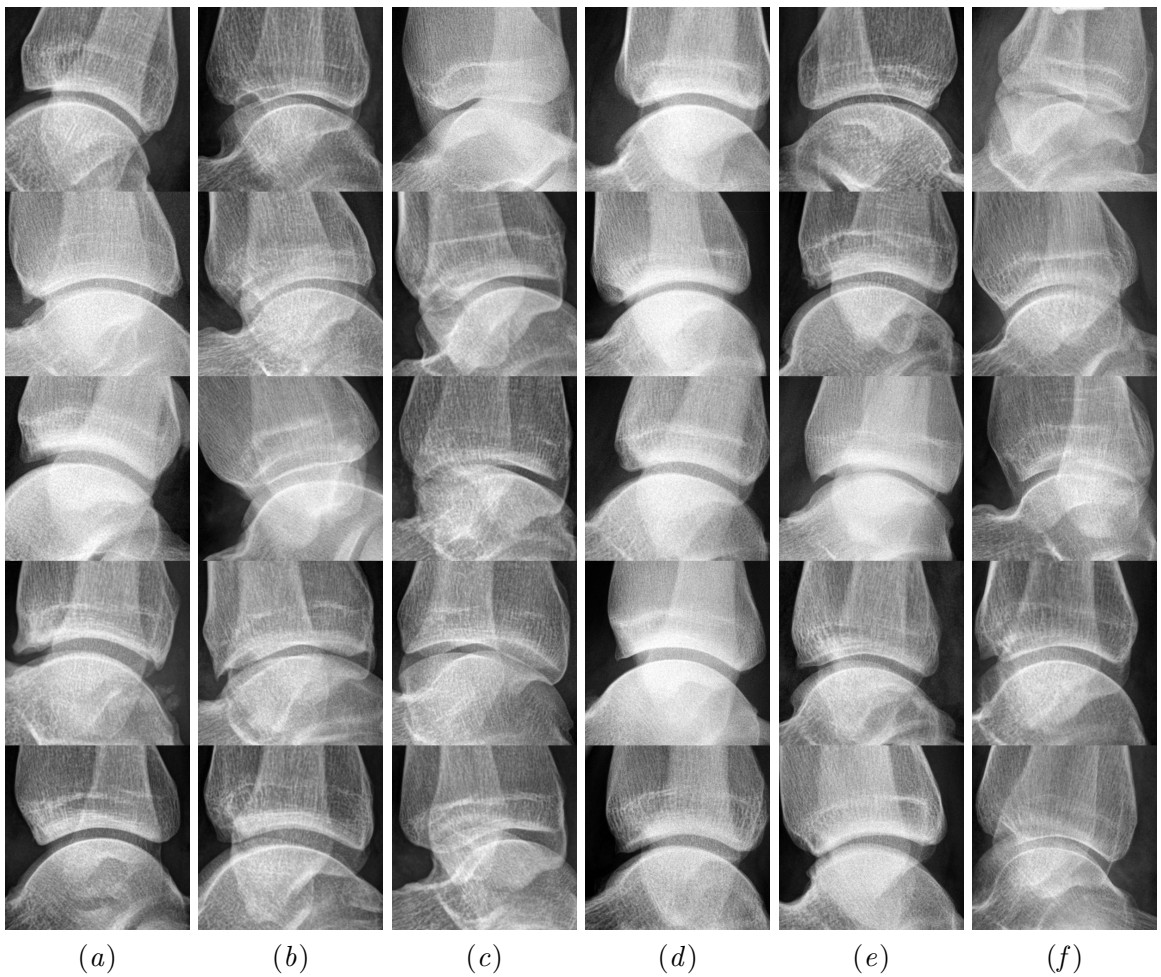

$(a)$ $(b)$ $(c)$ $(d)$ $(e)$ $(f)$

Figure 6: Each column shows five example ROIs of the labeled dataset in the *lateral* view with the expert label *1(a)*, *2(b)*, and *3(c)*; and five examples of unlabeled ROIs for which our framework predicts the quality classes *1(d)*, *2(e)*, and *3(f)*.

## Appendix B. Failure Cases

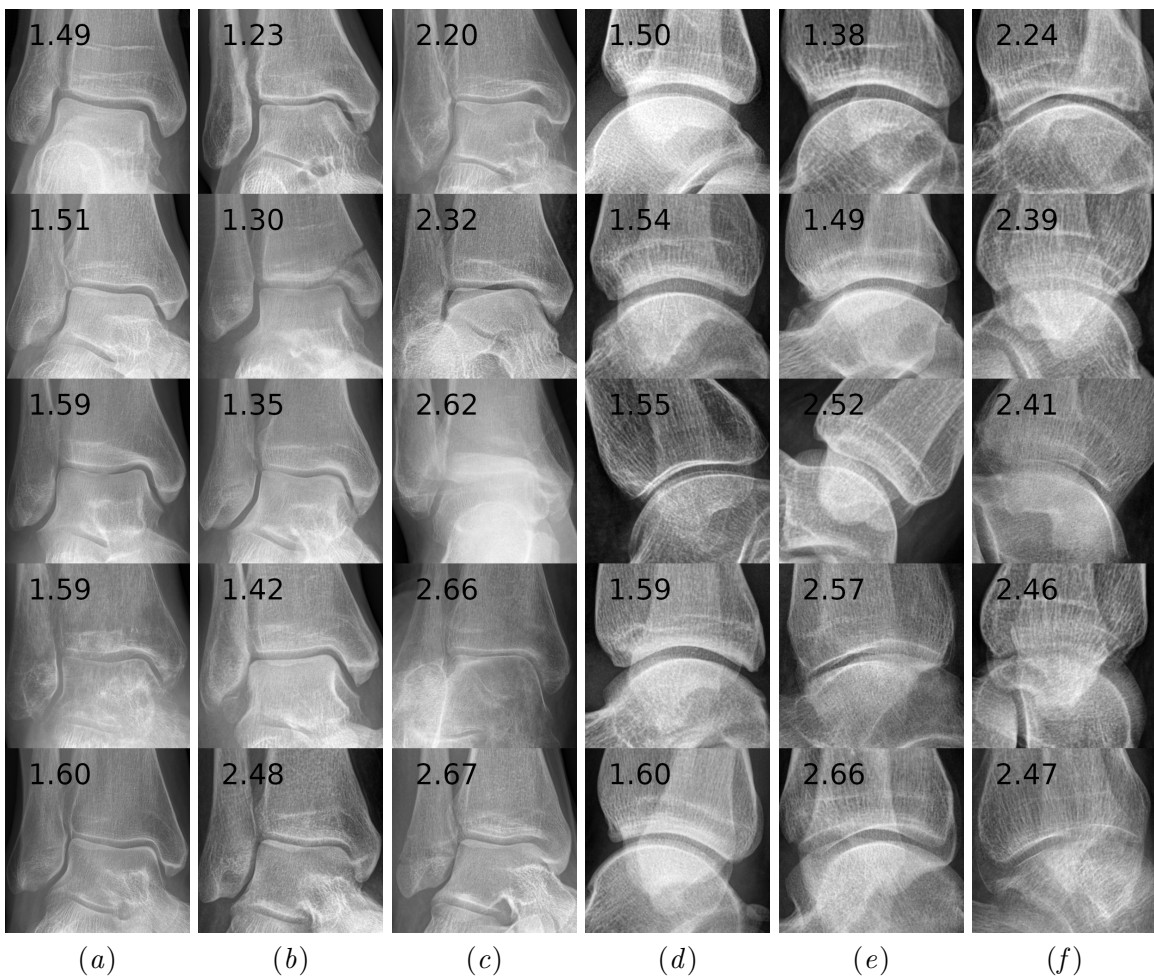

Figure 7: Each column shows five ROIs of the labeled dataset in the *anterior posterior* view with the expert label *1(a)*, *2(b)*, and *3(c)*; and five examples of ROIs in the *lateral* view with the expert label *1(d)*, *2(e)*, and *3(f)*. The quality assessed by our framework is printed on each ROI. For each class and view the five ROIs with the highest error between expert label and predicted quality are shown.

