# OpenReview forum: "An AI-based Framework for Diagnostic Quality Assessment of Ankle Radiographs"
_MIDL.io/2021/Conference — MIDL 2021_

### Official Review · AnonReviewer1 · 2021-03-01

**Confidence:** 5
**Preliminary Rating:** 4
**Recommendation:** Poster
**Final Rating:** 4

**Summary:**

This paper presents a method that classifies if ankle X-rays are of sufficient quality to read. They present a method which crops the ankle region and then classifies the patch. They compare the model's performance to radiologists and find their approach yields higher accuracy.
.........................

**Strengths:**

Excellent idea and execution.
Present a clear method.
Evaluation is sufficient to support their claims.
....................................................................................................

**Weaknesses:**

Dataset not public.
No discussion about failure cases.
.................................................................................................................................................

**Deanonymize Review:**

no

**Detailed Comments:**


This paper is great. The specific application and approach seems really like a winning application for AI in medicine. The paper clearly supports the utility of this method with their comparison to radiologists.

One concern with this study is that the method was not evaluated for out-of-distribution samples so it is not clear how robust it is compared to the radiologists. I think it is important to note this as a limitation of the claim you made.

Another thing I would like to see is the failure cases. Maybe you can add these to the appendix and add some discussion about why and where the method failed. Is there an intersection between the errors of the model and the radiologists?


**Final Rating Justification:**

No issues.

**Justification Of The Preliminary Rating:**

The paper is interesting, well written, and provides sufficient experimental evidence to support the claims.
.............................................................................................

**Paper Type:**

validation/application paper

**Questions To Address In The Rebuttal:**

Adding discussion and details of failure cases.


**Special Issue:**

no

---

> ### Author Response · Authors · 2021-03-17
> **Response to AnonReviewer1**
>
> Thank you for your helpful feedback. We will try to address your concerns as well as possible.
>
> > Dataset not public.
>
> - We do plan to make the dataset public, but we are still in the process of complying with privacy regulations and the EU data protection law.
>
>
> > One concern with this study is that the method was not evaluated for out-of-distribution samples so it is not clear how robust it is compared to the radiologists. I think it is important to note this as a limitation of the claim you made.
>
> - The model was trained on about 380 images and tested on the remaining ~95 images. Moreover, we evaluated the pipeline on the ~24000 images of the dataset described in 4.1, which was not labeled by experts. We have no accuracy for the predictions on this dataset, but the results were cross-checked by experts and examples are presented in Figure 5 and 6. Note that the examples in (d), (e) and (f) in these figures are from the unlabeled dataset of Section 4.1. Both datasets contain radiographs from five different X-ray machines. We added a sentence about the number of different X-ray machines to the paper.
>
>
> > Another thing I would like to see is the failure cases. Maybe you can add these to the appendix and add some discussion about why and where the method failed. Is there an intersection between the errors of the model and the radiologists?
>
> - Good point. We added examples of failure cases in the appendix in Figure 7. Note that there is no clear pattern that explains the deviation. We also added that sentence to the paper.

---

### Official Review · AnonReviewer4 · 2021-03-04

**Confidence:** 4
**Preliminary Rating:** 4
**Recommendation:** Best Paper Award
**Final Rating:** 4

**Summary:**

Quality assessment of radiological images is needed as images acquired might potentially be of insufficient quality to allow for diagnosis. However, systematic image acquisition is not desirable as it might unnecessarily expose the patient to radiation.

The abstract presents a framework for the automatic quality assessment of radiological images, which can either predict a rating or simply if the image is of sufficient quality or not given a radiological image. The proposed framework is split into three parts: View recognition, roi extraction and quality assessment. The first part classifies the image into one of two views. The second extracts a ROI to help the third part, which predicts either a quality rating or assess if the image is of sufficient quality.

The authors use two datasets, one novel, to train and assess the performance of their framework. Experimental results are given for each component of the proposed framework.

**Strengths:**

While the idea presented is in itself quite simple, it is nevertheless novel and allows for good results. The abstract is very well written: it is concise without cutting corners, it is well structured, everything is introduced where it should be, it is easy to understand, and the figure are plenty and helpful. The experiments are well defined, and the results are well reported.

**Weaknesses:**

Because the framework includes several components, it would have been interesting to see a justification for each component in the form of an ablation study on how each component affects the final quality assessment. What happens if the third component is reduced to a single network fed ROIs from both views ? Or even full images from both views ? While this question was briefly mentioned in section 5.3, a full ablation study would have benefit the paper.

Similarly, it would have been interesting to see comparison between the different components of the framework and their already-established individual counterparts. For example, while section 2 does mention that the proposed work has no direct "competitor":

> To our knowledge, there are no studies considering anatomical features for the diagnostic quality of radiographs in a deep-learning framework

, it would have been interesting to see how the proposed view detector performs against "classical" approaches [1,2] (both cited in section 2), even though they are indeed for chest images and not ankle images. It could also have been interesting to see the ROI extractor or the view detector performs against existing methods (for example [3], which is also cited in section 2).

[1]: Charles E. Willis, Thomas K. Nishino, Jered R. Wells, H. Asher Ai, Joshua M. Wilson, and Ehsan Samei. Automated quality control assessment of clinical chest images. Medical Physics, 45(10):4377–4391, October 2018. ISSN 2473-4209. doi: 10.1002/mp.13107.
[2]: Ehsan Samei, Yuan Lin, Kingshuk R. Choudhury, and H. Page McAdams. Automated characterization of perceptual quality of clinical chest radiographs: Validation and calibration to observer preference. Medical Physics, 41(11):111918, 2014. ISSN 2473-4209. doi: 10.1118/1.4899183.
[3]: Xiang Fang, Leah Harris, Wei Zhou, and Donglai Huo. Generalized Radiographic View Identification with Deep Learning. Journal of Digital Imaging, December 2020. ISSN 1618-727X. doi: 10.1007/s10278-020-00408-z.


**Deanonymize Review:**

no

**Detailed Comments:**

It is in the opinion of the reviewer that the results could have been better presented in a table summarizing the different results for the components instead of spread throughout the text.

**Final Rating Justification:**

The main weakness of the paper, namely its lack of ablation study, was addressed by the authors in a subsequent version. The paper is very well written, concise and complete and now more thorough in its experimentation.

**Justification Of The Preliminary Rating:**

This is an abstract of exceptional quality. I wish I could have written more in the "Strengths" section but as always, it is easier to express negative points than positives. The weaknesses presented are points that would be necessary in a full paper, but are absolutely not critical for an abstract. I am perfectly on board with accepting the paper in its current form.

**Paper Type:**

both

**Special Issue:**

yes

---

> ### Author Response · Authors · 2021-03-17
> **Response to AnonReviewer4**
>
> Thank you for the positive and constructive review. Regarding your concerns:
>
> > Because the framework includes several components, it would have been interesting to see a justification for each component in the form of an ablation study on how each component affects the final quality assessment. What happens if the third component is reduced to a single network fed ROIs from both views? Or even full images from both views ? While this question was briefly mentioned in section 5.3, a full ablation study would have benefit the paper.
>
> - Good suggestion. We did run that now and the results are included in the paper in Table 1 and are described in Section 5.3.
>
>
> > Similarly, it would have been interesting to see comparison between the different components of the framework and their already-established individual counterparts. It would have been interesting to see how the proposed view detector performs against "classical" approaches [1,2] (both cited in section 2), even though they are indeed for chest images and not ankle images. It could also have been interesting to see the ROI extractor or the view detector performs against existing methods (for example [3], which is also cited in section 2).
>
> - Since both the method and the scope of the ROI extraction in [1] and [2] is different, we do not see a good way to compare our results. They extract multiple small patches as ROIs along some predefined landmarks given by some external program, while we have always exactly one ROI per image.
>
> - While the approach of view classification in [3] is comparable, they have a different number of classes. When they classify into AP and LAT view, they also classify into oblique view, left and right. Because of that, a fair numerical comparison is hard to make.
>
>
> > It is in the opinion of the reviewer that the results could have been better presented in a table summarizing the different results for the components instead of spread throughout the text.
>
> - We added a table comparing to what improvement what step leads (Table 1) and a second one which summarizes the results for each step in the pipeline (Table 2).

---

> > ### Comment · AnonReviewer4 · 2021-03-20
> > **Improvements to readability**
> >
> > Thank you for addressing the comments made above. The inclusion of the two tables greatly improves the conciseness of the article. The authors rebuttal concerning comparison against prior work are sufficient. I have no further comment.

---

### Official Review · AnonReviewer3 · 2021-03-08

**Confidence:** 5
**Preliminary Rating:** 1
**Final Rating:** 1

**Summary:**

The authors proposed a deep learning pipeline to assess the quality of ankle radiographs. Instead of an end-to-end approach, the authors suggested a modular framework that combines view selection and ROI estimation to provide the final rating about the radiograph. The performance of the pipeline is validated in an in-house dataset.

**Strengths:**


1.	This is a clinically relevant problem that needs immediate attention.
2.	The authors put significant effort in creating the in-house dataset.
3.	The authors proposed a simple processing pipeline that makes sense and produce sensible results.



**Weaknesses:**

1.	Lack of quantitative evaluation as well as comparison
2.	Lack of methodological novelty
3.	Though main contribution of the paper is the annotated dataset, no clear intention from the authors about making it public.


**Deanonymize Review:**

no

**Detailed Comments:**

This is a fairly application-driven paper, which of course is welcome in MiDL. However, I expect to see a thorough evaluation of the method, including comparisons with baseline and state-of-the-art. Unfortunately none of this is present in the paper. This in combination with the lack of commitment in making the data public make this paper unsuitable to be presented in a conference of MiDL’s stature.

**Final Rating Justification:**

The authors are not interested in making the data public to accompany the paper. The methodological novelty question is dodged with an argument that "someone else is convinced". There are many ablation studies and simple baselines that can be performed for such a pipeline. The authors didn't even make a sincere effort towards that.

**Justification Of The Preliminary Rating:**

Basically the weaknesses are staggering. I don't see how the authors can improve this paper significantly during rebuttal. To re-iterate:
1.	Lack of quantitative evaluation as well as comparison
2.	Lack of methodological novelty
3.	Though main contribution of the paper is the annotated dataset, no clear intention from the authors about making it public.


**Paper Type:**

validation/application paper

**Questions To Address In The Rebuttal:**

None

**Special Issue:**

no

---

> ### Author Response · Authors · 2021-03-17
> **Response to AnonReviewer3**
>
> Thank you for your feedback and comments. We will try our best to answer your concerns:
>
> > Lack of quantitative evaluation as well as comparison
>
> - The model was trained on about 380 images and tested on the remaining ~95 images. Moreover, we evaluated the pipeline on the ~24000 images of the dataset described in 4.1, with was not labeled by experts. We have no accuracy for the predictions on this dataset, but the results were cross-checked by experts and examples are presented in Figure 5 and 6.
> - As we argue in Section 2 there is no related state-of-the-art approach, which is suited for a direct comparison. We added a comparison with an approach without the steps described in Section 3.1 and 3.2 as a baseline. The results are in Table 1.
>
>
> > Lack of methodological novelty
>
> - The methodological novelty is given by the way we combine different deep learning tools, as acknowledged by reviewer 4 “... the idea presented is in itself quite simple, it is nevertheless novel and allows for good results”.
>
>
> > Though main contribution of the paper is the annotated dataset, no clear intention from the authors about making it public
>
> - The main contribution is a solution for a novel application. Since the solution is based on deep learning, the data used for training is of course important.
> - We do plan to make the dataset public, but we are still in the process of complying with privacy regulations and the EU data protection law.

---

> ### Comment · AnonReviewer1 · 2021-03-18
> **Response from another reviewer**
>
> I voted to strongly accept this paper so I would like to discuss the negative points you raised and defend my perspective.
>
> > Lack of quantitative evaluation as well as comparison
>
> Section 5.3 has many quantitative evaluations. I believe they could be organized better but they compare to a baseline of radiologists and find "The mean accuracy over both views is 94.1% for the networks and 91.4% for the radiologists." which is a significant and interesting finding.
>
> > Lack of methodological novelty
>
> I find the specific use case they have put together to be novel. It is very challenging to find convincing use cases which can impact clinical pipelines. I believe this is one such use case and I have never heard it discussed before. Do you have a reference of this task being automated before?
>
> > Though main contribution of the paper is the annotated dataset, no clear intention from the authors about making it public.
>
> I don't believe this is the main contribution of the paper. I think the use case and the radiologist study is. And I think MIDL is a perfect fit for these contributions. In any case I find the private dataset an issue but the authors state it will be made public.

---

> > ### Comment · AnonReviewer3 · 2021-03-19
> > **Response to reviewer's comments**
> >
> > First of all, thank you for taking the time to discuss such an issue. I wholeheartedly support such discussion as this in turn makes us a better community.
> >
> > Let's start at the finish. The authors said they wish to make the dataset public at some point in future. While I do understand the reality of author's situation, I interpret that this paper won't accompany the public dataset. That confirms my initial suspicion.
> > Now I do appreciate your viewpoint that you do not believe that the dataset is the main contribution of the paper. But let's agree to disagree here.
> >
> > The authors dodged the methodological novelty question with an argument that "someone else is convinced". If you read my strength and detailed comment section, you'll see that I value the use-case as well. I just think MiDL is a more technical than clinical venue. So, if a paper lacks methodological novelty, it should complement it by thorough evaluation.
> >
> > This brings to my first concern. There are many ablation studies and simple baselines that can be performed for such a pipeline. And I am sure my peer-reviewer would agree such checks and balances are vital if you are developing a pipeline combining multiple deep neural networks. Unfortunately, the authors didn't even make a sincere effort to answer this concern.
> >
> > One red flag is fine with me. But, I can't recommend an acceptance with three red flags.

---

### Official Review · AnonReviewer2 · 2021-03-09

**Confidence:** 4
**Preliminary Rating:** 2

**Summary:**

Authors present a framework for diagnostic quality assessment of ankle radiographs. They use a combination of models that recognize the view of radiograph, extract the ROI and finally infer the quality, based on anatomical features. They show that based on such anatomical features, a modular deep-learning framework can serve as a quality control mechanism for the diagnostic quality of ankle radiographs.

**Strengths:**

+ Authors propose a modular deep-learning framework that can serve as a quality control mechanism for the diagnostic quality of ankle radiographs.
+ This work shows use of simple classification and segmentation Neural Networks to solve a larger task more efficiently

**Weaknesses:**

+ Why was a large model like _efficientnet-B0_ used for a simple task like view prediction? Based on image provided, even simple compute vision algorithms could have been used. Authors should compare their results to other non machine learning models. Given such small size of dataset, authors make no comment on the data size effects on results, specially given model used is so large.
+ Given ROIs are represented as rectangles in the ground truth data, its not clear if they are of fixed sizes for the two views? How tight they are?
+ When authors say "**When a single model was used to assess the quality for both radiographic views, the results were about 15 percentage points worse**", its not clear if the single model was trained on the combined views data, what was the sampling strategy when training such a combined view model?
+ Given raw ROIs are rectangular regions not true pixel-level segmentations, its not clear why do authors use a segmentation model like DeepLab V3?
+ Given the models were trained on the consensus of 4 radiologists, I am not sure if it is fair to compare against 1 vs consensus of 3 radiologists and claim that model is doing better than individual radiologists. It would be better to compare against data from different set of radiologists.

**Deanonymize Review:**

no

**Detailed Comments:**

+ Please clarify this statement: **This shows that the dataset is unevenly distributed across the classes, which was expected since most radiographs are neither perfect nor not diagnostic.**

**Justification Of The Preliminary Rating:**

This is an interesting work to showcase a combination of models to solve an interesting problem. However, for authors claims additional experiments are needed. Authors need to address issues pointed in the weaknesses section.

**Paper Type:**

validation/application paper

**Special Issue:**

no

---

> ### Author Response · Authors · 2021-03-17
> **Response to AnonReviewer2**
>
> Thank you for the detailed feedback. We would like to address your concerns as follows:
>
> > Why was a large model like efficientnet-B0 used for a simple task like view prediction? Based on image provided, even simple compute vision algorithms could have been used. Authors should compare their results to other non machine learning models. Given such small size of dataset, authors make no comment on the data size effects on results, specially given model used is so large.
>
> - The reason we used an efficientnet-B0 is, that we wanted to be able to apply the pipeline to other parts of the body without further modifications.
> - Regarding the size of the dataset, there might be a misunderstanding. For the training of the view prediction, we used 80% of the ~24000 images of the dataset described in Section 4.1, which should be sufficient. We made that clearer in the paper.
>
> > Given ROIs are represented as rectangles in the ground truth data, its not clear if they are of fixed sizes for the two views? How tight they are?
>
> - The labeled ROIs are always quadratic and of variable size, chosen to contain all the relevant anatomical features. Only the network of the step described in Section 3.3 (quality prediction) receives fixed ROIs resized to 224x224. We added a sentence to the end of Section 4.2 mentioning the variable size.
>
> > When authors say "When a single model was used to assess the quality for both radiographic views, the results were about 15 percentage points worse", its not clear if the single model was trained on the combined views data, what was the sampling strategy when training such a combined view model?
>
> - This sentence was indeed misleading. By introducing Table 1 and adding an explanation to Section 5.3, which also explains the sampling strategy, the training process should be clarified.
> When training a single model on both views, the data for AP and LAT is combined, and we sampled each view the same number of times.
>
> > Given raw ROIs are rectangular regions not true pixel-level segmentation, its not clear why do authors use a segmentation model like DeepLab V3?
>
> - Yes, one could argue that the segmentation network is misused for detection, but since we know that there is always exactly one instance of the ROI (always one ankle on each radiograph) the architecture is efficient.
>
> > Given the models were trained on the consensus of 4 radiologists, I am not sure if it is fair to compare against 1 vs consensus of 3 radiologists and claim that model is doing better than individual radiologists. It would be better to compare against data from different set of radiologists.
>
> - This is a valid point, we used the leave one out method because we did not have a different set of radiologists. Obviously our claims are bound to the way we evaluated and should be taken as a first estimate. We have added a sentence in Section 5.3 in order to clarify the claim.
>
> > Please clarify this statement: This shows that the dataset is unevenly distributed across the classes, which was expected since most radiographs are neither perfect nor not diagnostic.
>
> - The sentence was misleading and unnecessary, so we removed it.

---

### Author Response · Authors · 2021-03-17
**Response to Reviewers**

We thank the reviewers for their valuable comments and appreciate the positive remarks. We have uploaded an updated version of the paper and respond to the reviews individually in order to address all concerns as well as possible.

---

### Author Response · Authors · 2021-03-22
**Response to AnonReviewer3**

Reviewer 3 still raised three points in the Final Rating Justification, to which we would like to respond:

- Data: as we have already said, we do plan to make the dataset public, but are still in the process of complying with privacy regulations and the EU data protection law. This is not easy as it is patient data of several years.

- Novelty: the novelty lies in combining Deep Learning methods to solve a relevant but previously unsolved problem.

- Ablation study: like we said in our first response, but perhaps not clearly enough, we added an ablation study, included in Table 1 and described in Section 5.3..

---

### Meta-Review · Area_Chairs · 2021-03-31

**Recommendation:** Accept (Oral)

**Metareview:**

this paper certainly was considered very controversial. Two reviewers voted strong accept while one reviewer heavily opposed the work based on the perceived unwillingness to release data. One slightly negative reviewer did not revisit the paper after rebuttal. The positives clearly outweigh the negatives in this case and I would personally urge the authors to do their best to release data or at least code as it is within the spirit of MIDL (even though it's sometimes impossible and that shouldn't be reason for rejection)

**Paper Type:**

validation/application paper

---

### Decision · Program_Chairs · 2021-03-31

Accept